# Bed-side measures for diagnosis of low muscle mass, sarcopenia, obesity, and sarcopenic obesity in patients with chronic kidney disease under non-dialysis-dependent, dialysis dependent and kidney transplant therapy

**Natália Tomborelli Bellafronte**[1]*, **Gabriel Ruiz Sizoto**[2], **Lorena Vega-Piris**[3], **Paula Garcia Chiarello**[4☯], **Guillermina Barril Cuadrado**[5☯]

1 Post-Graduate Program in Health Sciences, Ribeirão Preto Faculty of Medicine, University of São Paulo, Ribeirão Preto, São Paulo, Brazil, 2 Nutrition and Metabolism Undergraduate Course, Ribeirão Preto Faculty of Medicine, University of São Paulo, Ribeirão Preto, São Paulo, Brazil, 3 Methodology Unit, Instituto de Investigación Sanitaria del Hospital Universitario de la Princesa, Madrid, Spain, 4 Department of Health Sciences, Ribeirão Preto Faculty of Medicine, University of São Paulo, Ribeirão Preto, São Paulo, Brazil, 5 Nephrology Department, Hospital Universitario La Princesa, Madrid, Spain

☯ These authors contributed equally to this work.
* natalia.bellafronte@usp.br

## Abstract

Muscle depletion and sarcopenic obesity are related to a higher morbimortality risk in chronic kidney disease (CKD). We evaluated bed-side measures/indexes associated with low muscle mass, sarcopenia, obesity, and sarcopenic obesity in CKD and proposed cutoffs for each parameter. Sarcopenia was diagnosed according to the European Working Group on Sarcopenia in Older People revised consensus applying dual energy X-ray absorptiometry (DXA) and hand grip strength (HGS), and obesity according to the International Society for Clinical Densitometry. Anthropometric parameters including calf (CC) and waist (WC) circumferences and WC/height (WC/H); bioelectrical impedance data including appendicular fat free mass (AFFM) and fat mass index (FMI) were assessed. ROC analysis and area under the curve (AUC) were applied for performance analyses. AFFM and CC presented the best performances for low muscle mass diagnosis–AFFM AUC for women was 0.96 and for men, 0.94, and CC AUC for women was 0.89 and for men, 0.85. FMI and WC/H were the best parameters for obesity diagnosis–FMI AUC for women was 0.99 and for men, 0.96, and WC/H AUC for women was 0.94 and for men, 0.95. The cutoffs (sensibility and specificity, respectively) for women were AFFM≤15.87 (90%; 96%), CC≤35.5 (76%; 94%), FMI>12.58 (100%; 93%), and WC/H>0.66 (91%; 84%); and for men, AFFM≤21.43 (98%; 84%), CC≤37 (88%; 69%), FMI>8.82 (93%; 88%), and WC/H>0.60 (95%; 80%). Sensibility and specificity for sarcopenia diagnosis were for AFFM+HGS in women 85% and 99% and in men, 100% and 99%; for CC+HGS in women 85% and 99% and in men, 100% and 100%; and for sarcopenic obesity were for FMI+AFFM in women 75% and 97% and in men, 75% and 95%. The tested bed-side measures/indexes presented excellent performance.

**Data Availability Statement:** All relevant data are within the manuscript and its Supporting Information files.

**Funding:** This research was funded by "Coordenação de aperfeiçoamento de pessoal de nível superior (CAPES)" (N.T.B., grant number 001), URL: <https://www.capes.gov.br/>. The research was also funded by "Fundação de amparo à pesquisa do estado de São Paulo (FAPESP)" (G. R. S., grant number 26305-9), URL:<https://www.fapesp.br/>. The funders had no role in study design, data collection and analysis, decision to publish, or preparation of the manuscript.

**Competing interests:** The authors have declared that no competing interests exist.

**Abbreviations:** ABSI, a body shape index; AFFM, appendicular fat free mass; ALM, appendicular lean mass; ALMI, appendicular lean mass index; AMA, arm muscle area; APTM, adductor pollicis muscle thickness; AUC, area under the curve; BCM, body cell mass; BCMI, body cell mass index; BMI, body mass index; BIS, bioelectrical impedance spectroscopy; cAMA, corrected arm muscle area; CC, calf circumference; CKD, chronic kidney disease; DXA, dual energy X-ray absorptiometry; ECW/ICW, extra to intracellular water ratio; eGFR, estimated glomerular filtration rate; FFMBCM, fat free mass by body composition monitor; FFMIBCM, fat free mass index by body composition monitor; FMBCM, fat mass body composition monitor; FMIBCM, fat mass index body composition monitor; FMtr, fat mass of trunk; HD, hemodialysis; HGS, hand grip strength; ICW, intracellular water; KTx, kidney transplant; LM, lean mass; LMI, lean mass index; MAC, mid-arm circumference; MAMC, mid-arm muscle circumference; NDD, non-dialysis dependent; OH, overhydration; OR, odds ratio; PD, peritoneal dialysis; ROC, receiver operator characteristic; pFFM, predicted fat free mass; pFFMI, predicted fat free mass index; pFM, predicted fat mass; pFMI, predicted fat mass index; PhA, phase angle; R, resistance; TSF, triceps skin fold thickness; WC, waist circumference; WC/H, waist circumference for high ratio; Xc, reactance.

# Introduction

The global increase in the prevalence of diabetes mellitus, hypertension, obesity and aging has shaped chronic kidney disease (CKD) epidemiology, increasing its incidence and prevalence [1, 2]. Also, from 1990 to 2016, CKD went from the 18th to the 12th leading cause of death [3], demonstrating a lack of progress in disease management.

Metabolic disorders present in CKD, such as uremic toxins accumulation, chronic inflammation, metabolic acidosis, oxidative stress, hormonal imbalance, and cellular metabolism disorders, increase skeletal muscle catabolism and decrease muscle regeneration [4, 5]. Protein catabolism is worsened by other typical conditions in CKD, such as diet restrictions, disturbances in appetite-regulating hormones, uremia-related gastrointestinal symptoms, physical inactivity, nutrient malabsorption, and nutrient loss into the dialysate [4, 5]. Therefore, muscle impairment is frequent among CKD patients [6–8] and is related to adverse outcomes [6, 7, 9]. Sarcopenia (presence of low muscle mass and strength [10]) is also common in CKD, and was proved to seriously worsen clinical prognosis, decrease quality of life, and increase mortality risk [6]. Hand grip strength (HGS) and dual energy X-ray absorptiometry (DXA) are the recommended methods for sarcopenia screening [10].

In addition to muscle depletion, obesity is also common among CKD patients [10, 11]. Obesity has a controversial role in survival rates and clinical prognosis in CKD patients [6, 9, 12]. Currently, body mass index (BMI) is the most widely used obesity measure, but the index is an imperfect measure of adiposity [10]. However, according to the International Society for Clinical Densitometry [13], obesity is more reliably defined as high adiposity evaluated by DXA. Sarcopenic obesity, a combination of high body fat with muscle depletion, also affects CKD patients and strongly contributes to a worse clinical status compared with either of the two conditions alone [14].

As negative changes of body composition and nutritional status significantly increase morbidity and mortality risk in CKD patients, the early diagnosis of such changes is of fundamental importance. However, DXA availability is restricted, usually applied in diagnostic studies and rarely feasible in clinical practice. Therefore, the use of tools that are more easily available in routine nutritional assessment, such as anthropometry and bioelectrical impedance analyses, could help in the early identification of nutritional status impairment, improving clinical outcomes by early interventions.

Thus, the aim of this study was to evaluate the diagnostic ability of bed-side measures of muscle mass and adiposity and propose diagnostic cutoffs for low muscle mass, sarcopenia, obesity, and sarcopenic obesity in adult CKD patients on non-dialyses-dependent (NDD), hemodialysis (HD), peritoneal dialysis (PD), and kidney transplant (KTx) treatment. We further evaluated the agreement and correlation between bed-side measures and DXA and their relationship with diagnosed conditions.

# Materials and methods

## Study population

This was an observational cross-sectional study followed by a prospective analysis. Subjects (18 to 60 years old) were recruited from May 2017 to May 2019 at a tertiary care hospital, the University Hospital of the Ribeirão Preto Medical School and at a dialysis clinic, the Nephrology Service of Ribeirão Preto, as follows: CKD patients in NDD treatment in stages 3b to 5; patients in dialysis for at least 3 months, with HD by a 4-hour dialysis session, 3 times per week, through an arteriovenous fistula, and in PD without peritonitis for the last 30 days; patients with KTx for at least 6 months with CKD stages 1 to 3a. Inpatients and patients that were wheelchair users, had a body weight above 140 kg or BMI higher than 40 kg/m$^2$, with

amputations or electronic implant, acute infections, cancer diagnosis, acquired immunodeficiency syndrome, and other conditions that could alter body composition were excluded.

Sample size was calculated based on sarcopenia prevalence in CKD, assuming an expected prevalence of 5%, with estimation of at least 80 patients for each CKD treatment [15].

Residual renal function was estimated by measuring mean urinary creatinine clearances adjusted for $1.73m^2$ from 24-hour urine collection performed from 1 to 7 days before study assessment. Kt/V was estimated by weekly clearance of dialyzed urea adjusted by total body water in the week prior to the study.

The Ribeirão Preto Medical School Ethics Committee approved the study (protocol number 2053045). All patients were invited by the researcher and those interested in participating the study read and signed the informed consent form before the procedures began.

Clinical and biochemical data were collected from medical records up to 10 days prior to assessment. Anthropometric (weight with precision of 0.1 kg, height and circumferences with precision of 0.1 cm), multifrequency bioelectrical impedance spectroscopy (BIS), and DXA measurements were performed consecutively at the same visit by a single trained professional after an 8-hour fast, empty urinary bladder, drainage of the peritoneal dialysate, just after the midweek hemodialysis session, with patients wearing light clothes, without shoes and on the right side of the body (except if a fistula was present).

## Body composition, anthropometric, and functional assessment

DXA (Hologic Discovery A®, USA) was performed with segmental evaluation of both arms, both legs, and trunk and total mass was calculated by adding the values of segments [13]. DXA provided data of lean mass (LM) and its index (LMI), appendicular lean mass (ALM) and its index (ALMI), fat mass (FM) and its index (FMI), LM/FM ratio, and fat mass of the trunk region (FMtr). A whole-body scan was carried after a daily calibration of the device by scanning a spine phantom. ALM was determined by adding LM of both arms and legs. ALMI, LMI, and FMI were calculated as the respective mass value normalized for squared height.

BIS (BCM®, Fresenius Medical Care, DEU) analysis was applied in a tetra-polar unilateral whole-body wrist-to-ankle protocol [16] after a 10-minute adaptation in a supine position. Phase angle [17], intracellular water, over-hydration, body cell mass and its index, fat-free mass (FFMBCM) and its index (FFMIBCM), and fat mass (FMBCM) and its index (FMIBCM) were estimated by the equipment's prediction formulas [18, 19]. Appendicular fat free mass (AFFM) was calculated by applying Sergi equation [20], as advised by the revised consensus on sarcopenia of the European Working Group on Sarcopenia in Older People [21]. Two new equations recently develop by our research group [22] were applied for FFM (pFFM) and its index (pFFMI), and FM (pFM) and its index (pFMI). Sex was coded 1 for men and 0 for women, age was recorded in years, BMI in kg/m², waist circumference (WC) in cm, resistance (R) and reactance (Xc) of 50 frequency in ohms, extra to intracellular water ratio (ECW/ICW) in L, and FMBCM and FFMBCM in kg using the formulas:

$$FFM(kg) = e^{(1.26010+0.08458*sex+0.00019*age+0.01131*BMI+0.00307*WC+0.00023*R+0.00242*Xc+0.90554*ECW/ICW+0.02039*FFMBCM)}$$

$$FM(kg) = 31.79970 - 2.92130 * sex - 0.02109 * age - 0.18604 * BMI + 0.04547 * WC \\ - 0.01221 * R - 0.06439 * Xc - 19.17652 * ECW/ICW + 0.86347 * FMBCM$$

A flexible non-stretchable plastic tape was used to measure circumferences. WC was obtained at umbilical scar level [23] and normalized for height in centimeters (WC/H). Calf circumference (CC) was measured with subjects seating down, knees at 90˚ and at the calf greatest circumference [24]. Mid-arm circumference and triceps skin fold thickness were

measured at mid-point between olecranon and acromion [25]. Triceps skin fold thickness was assessed with an adipometer (Lange®, Cambridge Scientific Industries, Inc); three measures were taken and their mean was used.

Mid arm muscle circumference, arm muscle area, corrected arm muscle area [25], a body shape index [26], and conicity index [27] were calculated according to previous studies.

The adductor pollicis muscle thickness was measured by pinching with the adipometer at the vertex of the imaginary triangle formed by the thumb and index fingers, three times, with the average used as the result [28].

HGS was evaluated by a pneumatic dynamometer (Charder®, MG 4800) with subjects seated and asked to grip as hard as possible for three times. The highest value was recorded [29].

## Diagnosis of pre-sarcopenia, low muscle mass, sarcopenia, obesity and sarcopenic obesity

Diagnoses of pre-sarcopenia, low muscle mass, and sarcopenia were established according to the new consensus on sarcopenia of the European Working Group on Sarcopenia in Older People [21]: pre-sarcopenia as HGS<16 kg for women and HGS<27 kg for men; low muscle mass as ALM<15 kg for women and ALM<20 kg for men, assessed by DXA; and sarcopenia as concomitant presence of pre-sarcopenia and low muscle mass.

Obesity was diagnosed according to the recommendation of the International Society for Clinical Densitometry [13] as FMI>13 kg/m$^2$ for women and FMI>9 kg/m$^2$ for men assessed by DXA [30].

Sarcopenic obesity was defined as the concomitant presence of low muscle mass and obesity.

## Statistical analysis

Data normality was assessed using the Shapiro-Wilk test. Continuous variables are reported as means ± SDs and categorical variables as frequencies and percentages. Comparisons between groups were made by unpaired Student's t-test or ANOVA with Bonferroni post-test for continuous variables. For categorical variables, the Chi-squared test was applied.

ROC curves were designed for each bed-side measure/index of muscle or adiposity with the use of DXA predictors of low muscle mass (ALM) and obesity (FMI), respectively, to identify sensitivity and specificity. The determination of the cutoffs was based on the values that maximized simultaneously sensitivity and specificity.

Logistic regression analysis was performed to estimate the OR of clinical, anthropometric, bioelectrical impedance, and body composition data for the official diagnosis of low muscle mass and obesity.

Agreement between official and bed-side diagnosis was evaluated by Cohen's kappa coefficient. Pearson's correlation was performed to asses associations between bed-side and DXA measures.

Analyses were carried out with the SPSS Statistics 23 (IBM, NY, USA). A 5% significance level was considered in all analysis.

# Results

## Study population

We evaluated 265 patients with a mean age of 48±10 years, 51% (n = 136) men, 31% in NDD (men, n = 46; women, n = 37), 29% in HD (men, n = 34; women, n = 44), 9% in PD (men,

n = 8; women, n = 15) and 31% in KTx (men, n = 48; women, n = 33) treatment. CKD was secondary to systemic arterial hypertension in 77 patients, glomerulonephritis in 66, diabetes mellitus in 27, glomerulosclerosis and systemic syndromes in 18, polycystic kidney in 20, and unknown cause in 57. Estimated glomerular filtration rate was $18.62\pm8.52$ and $70.37\pm18.39$ mL/min/1.73m$^2$ for NDD and KTx treatment patients, respectively. Kt/V was $1.68\pm0.58$ and $2.59\pm0.51$ for HD and PD patients, respectively. Duration of treatment was $75\pm62$, $15\pm17$, and $92\pm61$ months for HD, PD, and KTx patients, respectively.

Compared to men, women had significantly lower ALM ($14\pm3$ *vs* $22\pm4$ kg), LM ($34\pm6$ *vs* $47\pm8$ kg), ALMI ($5.9\pm1.00$ *vs* $7.6\pm1.13$ kg/m$^2$), LMI ($13.6\pm2.12$ *vs* $16.5\pm2.40$ kg/m$^2$), LM/FM ratio ($1.5\pm0.4$ *vs* $2.5\pm1.15$), HGS ($21\pm5$ *vs* $39\pm8$ kg), phase angle ($5.5\pm0.93$ *vs* $6.3\pm0.91˚$) and extracellular water ($13\pm2.4$ *vs* $18\pm3.2$ L), and had higher FM ($25\pm8$ *vs* $22\pm8$ kg) and FMI ($10\pm3.4$ *vs* $7.61\pm2.9$ kg/m$^2$).

PD patients were younger (NDD, $48\pm10$; HD, $47\pm10$; PD, $40\pm12$; KTx, $49\pm8$ years old, $p\leq0.05$). NDD patients had higher LM (NDD, $44\pm10$; HD, $38\pm9$; PD, $37\pm10$; KTx, $40\pm9$ kg, $p\leq0.05$), LMI (NDD, $16.2\pm3.0$; HD, $14.3\pm2.10$; PD, $13.7\pm2.34$; KTx, $15.0\pm2.43$ kg/m$^2$, $p\leq0.05$), and extracellular water (NDD, $17\pm3.8$; HD, $14\pm3$; PD, $15\pm3$; KTx, $15\pm3$ L, $p\leq0.05$). ALM (NDD, $20\pm5$; HD, $17\pm4$; PD, $18\pm6$; KTx, $18\pm5$ kg, $p\leq0.05$) and FM (NDD, $25\pm9$; HD, $21\pm9$; PD, 21.6; KTx, $23\pm7$ kg, $p\leq0.05$) were different only between NDD and HD groups. ALMI (NDD, $7.3\pm1.49$; HD, $6.3\pm1.13$; PD, $6.4\pm1.50$; KTx, $6.8\pm1.29$ kg/m$^2$, $p\leq0.05$) was different only between NDD and HD, or between NDD and PD patients; HGS (NDD, $31\pm12$; HD, $27\pm10$; PD, $30\pm13$; KTx, $33\pm11$ kg, $p\leq0.05$) was different only between HD and KTx groups. For FMI (NDD, $9.3\pm3.50$; HD, $8.3\pm3.79$; PD, $7.9\pm2.44$; KTx, $8.9\pm2.93$ kg/m$^2$) and phase angle (NDD, $5.9\pm0.97$; HD, $5.8\pm1.17$; PD, $5.8\pm0.93$; KTx, $5.9\pm0.83˚$) there was no difference between CKD groups ($p>0.05$).

## Official diagnosis of pre-sarcopenia, low muscle mass, sarcopenia, obesity, and sarcopenic obesity

Prevalence of low muscle mass (63 *vs* 37%) and sarcopenia (10 *vs* 4%) were higher and prevalence of obesity (18 *vs* 30%) was lower in women than men ($p\leq0.05$). Prevalences of pre-sarcopenia (15 *vs* 7%, respectively) and sarcopenic obesity (6 *vs* 6%, respectively) were not different between women and men ($p>0.05$).

Pre-sarcopenia was present in 14 (n = 12), 13 (n = 10), 13 (n = 3), and 4% (n = 3) of NDD, HD, PD, and KTx patients, respectively. Low muscle mass was present in 28 (n = 23), 69 (n = 54), 52 (n = 12), and 52% (n = 42) of NDD, HD, PD, and KTx patients, respectively, with difference among groups ($p\leq0.05$). Sarcopenia was present in 5 (n = 4), 12 (n = 9), 9 (n = 2), and 4% (n = 3) of NDD, HD, PD and KTx patients, respectively. Obesity was present in 35 (n = 29), 17 (n = 13), 4 (n = 1), and 26% (n = 21) of NDD, HD, PD, and KTx patients, respectively, with difference among groups ($p\leq0.05$). Sarcopenic obesity was present in 1 (n = 1), 10 (n = 8), and 9% (n = 7) of NDD, HD, and KTx patients, respectively, with difference among groups ($p\leq0.05$).

In Table 1, data about subgroups with and without the studied conditions, stratified by sex and with comparisons between groups are presented. In general, low muscle mass and sarcopenic patients presented lower values of muscle and adiposity compared to patients without the condition. Muscle and adiposity were higher for obese than non-obese subjects. Sarcopenic obese patients presented higher adiposity and lower muscle measurements compared to non-sarcopenic obese patients.

Adiposity and muscle mass were protective factors for low muscle mass; phase angle was a protective factor only for women (Table 2). Diabetes mellitus and high total and central adiposity, mainly, were risk factors for obesity (Table 2).

**Table 1. Descriptive data of official diagnostic of pre-sarcopenia, low muscle mass, obesity, sarcopenia and sarcopenic obesity stratified by sex.**

| Variables | Pre-Sarcopenia | | Low Muscle Mass | | Obesity | | Sarcopenia | | Sarcopenic Obesity | |
|---|---|---|---|---|---|---|---|---|---|---|
| | Present | Absent | Present | Absent | Present | Absent | Present | Absent | Present | Absent |
| | $\bar{X} \pm SD$ | $\bar{X} \pm SD$ | $\bar{X} \pm SD$ | $\bar{X} \pm SD$ | $\bar{X} \pm SD$ | $\bar{X} \pm SD$ | $\bar{X} \pm SD$ | $\bar{X} \pm SD$ | $\bar{X} \pm SD$ | $\bar{X} \pm SD$ |
| *Female sample* | | | | | | | | | | |
| n | 19 | 110 | 81 | 48 | 23 | 106 | 13 | 43 | 8 | 33 |
| Age (years) | 49±9 | 47±10 | 47±10 | 48±9 | **53±6**\* | **47±10**\* | 49±10 | 48±9 | 52±5 | 46±9 |
| Dialysis/kidney transplant time (month) | 75±51 | 70±62 | 76±68 | 66±44 | 78±60 | 72±63 | 79±51 | 67±44 | 75±72 | 61±45 |
| eGFR (ml/min/1.73m$^2$) | 35.34 ±38.69 | 42.29 ±29.01 | **50.31 ±31.30**\* | **32.02 ±25.29**\* | 39.01 ±27.59 | 42.34 ±30.69 | 57.20 ±46.83 | 34.58 ±25.93 | **79.43 ±29.28**\* | **33.55 ±29.63**\* |
| KT/V | 1.96±0.56 | 2.02±0.78 | 2.04±0.79 | 1.93±0.59 | 1.76±0.30 | 2.04±0.78 | 1.94±0.59 | 1.91±0.61 | 1.89±0.17 | 2.01±0.59 |
| Weight (kg) | 61±14 | 65±13 | **58±9**\* | **77±10**\* | **80±9**\* | **61±12**\* | **56±11**\* | **77±10**\* | 72±6 | 73±9 |
| Body mass index (kg/m$^2$) | 26±5.5 | 26±5.1 | **24±4.1**\* | **30±4.4**\* | **34±2.9**\* | **25±4.0**\* | **24±4.6**\* | **30±4.5**\* | **32±2.9**\* | **28±3.4**\* |
| Hand grip strength (kg) | **13.4±1.8**\* | **22.7±4.6**\* | **20.0±4.4**\*\* | **23.6±6.2**\* | 19.6±4.6 | 21.8±5.5 | **14.0±1.0**\* | **25.0±4.9**\* | **16.5±1.4**\* | **24.8±6.5**\* |
| Phase angle (°) | 5.33±1.40 | 5.58±0.85 | **5.41±0.98**\* | **5.77±0.82**\* | 5.51±0.70 | 5.55±0.98 | 5.32±1.55 | 5.82±0.79 | 5.51±0.78 | 5.88±0.85 |
| Over-hydration (L) | -0.11±1.79 | -0.22±1.29 | -0.20±1.33 | -0.21±1.42 | **-0.71 ±1.33**\* | **-0.10 ±1.35**\* | -0.35±1.78 | -0.30±1.36 | **-1.40±0.91**\* | **-0.15±1.44**\* |
| Appendicular lean mass (kg) | 13.5±2.9 | 14.7±2.9 | **12.7±1.4**\* | **17.5±2.2**\* | **15.8±2.5**\* | **14.2±2.9**\* | **12.1±1.6**\* | **17.5±2.2**\* | **13.1±0.9**\* | **17.6±2.4**\* |
| Lean mass (kg) | 31.7±6.8 | 33.9±6.1 | **30.0±3.3**\* | **39.7±5.1**\* | **36.6±5.6**\* | **32.9±6.2**\* | **28.7±4.3**\* | **39.7±5.1**\* | **30.7±2.1**\* | **39.7±5.6**\* |
| Trunk fat mass (kg) | 12.6±4.7 | 12.8±4.7 | **10.9±3.9**\* | **15.8±4.4**\* | **19.4±3.1**\* | **11.3±3.6**\* | **11.1±4.4**\* | **15.7±4.5**\* | **18.1±3.0**\* | 13.8±3.4 |
| Fat mass (kg) | 23.4±8.4 | 24.8±8.2 | **21.4±7.0**\* | **30.1±7.3**\* | **36.9±3.9**\* | **22.0±6.3**\* | **21.2±7.9**\* | **30.2±7.4**\* | **35.4±4.0**\* | **26.6±5.7**\* |
| Appendicular lean mass index (kg/m$^2$) | 5.8±1.1 | 5.9±1.0 | **5.3±0.6**\* | **6.8±0.8**\* | **6.6±0.9**\* | **5.7±0.9**\* | **5.3±0.7**\* | **6.8±0.8**\* | **5.7±0.4**\* | **6.7±0.8**\* |
| Lean mass index (kg/m$^2$) | 13.6±2.3 | 13.6±2.1 | **12.5±1.3**\* | **15.5±1.9**\* | **15.3±1.9**\* | **13.2±1.9**\* | **12.5±1.5**\* | **15.4±1.9**\* | **13.5±0.9**\* | **15.1±1.9**\* |
| Fat mass index (kg/m$^2$) | 10.1±3.7 | 9.9±3.4 | **8.9±3.1**\* | **11.8±3.1**\* | **15.5±1.4**\* | **8.9±2.4**\* | **9.3±3.6**\* | **11.8±3.1**\* | **15.5±1.9**\* | **10.2±2.2**\* |
| Lean mass/Fat mass | 1.5±0.55 | 1.5±0.48 | 1.5±0.50 | 1.4±0.46 | **1.00±0.1**\* | **1.6±0.47**\* | 1.5±0.62 | 1.4±0.47 | **0.9±0.06**\* | **1.6±0.47**\* |
| *Male sample* | | | | | | | | | | |
| n | 11 | 127 | 51 | 86 | 51 | 96 | 5 | 81 | 11 | 53 |
| Age (years) | 48±10 | 47±10 | 48±11 | 47±10 | 49±9 | 47±11 | 48±10 | 47±10 | **52±5**\* | **46±11**\* |
| Dialysis/kidney transplant time (month) | 125±61 | 74±62 | 75±67 | 79±60 | 63±43 | 81±68 | 142±62 | 78±60 | 57±54 | 85±68 |
| eGFR (ml/min/1.73m$^2$) | 34.74 ±34.23 | 46.71 ±29.05 | **55.65 ±28.84**\* | **41.44 ±28.82**\* | 42.09 ±29.21 | 47.94 ±29.58 | 48.97 ±51.90 | 42.58 ±29.24 | 57.12 ±28.67 | 41.79 ±28.50 |
| KT/V | 1.63±0.22 | 1.68±0.55 | 1.67±0.51 | 1.67±0.56 | 1.66±0.74 | 1.67±0.48 | 1.75±0.13 | 1.71±0.58 | 1.41±0.08 | 1.61±0.41 |
| Weight (kg) | 72±14 | 78±15 | **64±8**\* | **84±13**\* | **91±10**\* | **71±12**\* | **62±12**\* | **85±13**\* | 76±3 | 78±11 |
| Body mass index (kg/m$^2$) | 27±4.1 | 27±4.8 | **24±3.3**\* | **29±4.5**\* | **32±3.0**\* | **25±3.3**\* | **24±2.3**\* | **29±4.5**\* | **29±2.1**\* | **26±3.1**\* |
| Hand grip strength (kg) | **22.4±3.2**\* | **40.6±7.1**\* | **35.7±6.7**\* | **41.3±8.6**\* | 38.3±7.8 | 39.7±8.7 | **20.9±3.8**\* | **42.4±7.4**\* | **35.2±5.0**\* | **42.7±8.7**\* |
| Phase angle (°) | 5.52±1.28 | 6.34±0.86 | 6.20±0.89 | 6.32±0.94 | 6.29±0.80 | 6.27±0.97 | **5.25 ±0.93**\* | **6.35 ±0.89**\* | 6.30±0.69 | 6.33±1.01 |
| Over-hydration (L) | 1.39±2.46 | 0.45±1.66 | **0.11±1.40**\* | **0.74±1.87**\* | 0.32±1.68 | 0.59±1.76 | 1.36±1.99 | 0.71±1.80 | -0.51±1.73 | 0.90±2.02 |
| Appendicular lean mass (kg) | **19.1±3.0**\* | **22.2±3.9**\* | **17.9±1.6**\* | **24.3±2.8**\* | **23.5±3.8**\* | **21.3±3.8**\* | **16.8±2.5**\* | **24.5±2.8**\* | **17.9±0.9**\* | **23.9±2.7**\* |
| Lean mass (kg) | 42.7±6.9 | 47.6±7.9 | **39.4±3.8**\* | **51.7±6.0**\* | **51.3±7.6**\* | **45.4±7.5**\* | **37.1±5.2**\* | **52.0±6.1**\* | **40.8±2.5**\* | **50.4±5.8**\* |
| Trunk fat mass (kg) | 12.5±4.5 | 12.5±5.2 | **10.1±4.3**\* | **13.7±5.2**\* | **18.0±2.9**\* | **9.9±3.8**\* | 10.4±4.7 | 13.8±5.2 | **16.3±2.7**\* | **10.8±3.9**\* |
| Fat mass (kg) | 22.1±7.2 | 21.8±8.3 | **17.8±6.6**\* | **24.1±8.2**\* | **30.8±4.6**\* | **17.9±6.0**\* | 18.3±6.7 | 24.2±8.2 | **27.7±3.4**\* | **19.4±6.3**\* |
| Appendicular lean mass index (kg/m$^2$) | 7.1±0.9 | 7.7±1.1 | **6.6±0.6**\* | **8.3±0.9**\* | **8.3±1.1**\* | **7.3±1.0**\* | **6.4±0.6**\* | **8.3±0.9**\* | **6.9±0.6**\* | **8.0±0.8**\* |
| Lean mass index (kg/m$^2$) | 15.8±2.0 | 16.5±2.4 | **14.4±1.5**\* | **17.6±2.0**\* | **18.2±2.1**\* | **15.7±2.1**\* | **14.2±1.1**\* | **17.7±2.0**\* | **15.6±1.2**\* | **16.9±1.7**\* |
| Fat mass index (kg/m$^2$) | 8.2±2.4 | 7.6±2.9 | **6.5±2.5**\* | **8.2±2.9**\* | **11.0±1.6**\* | **6.1±1.9**\* | 6.9±1.9 | 8.2±2.9 | **10.6±1.5**\* | **6.5±1.9**\* |

(*Continued*)

**Table 1.** (Continued)

| Variables | Pre-Sarcopenia | | Low Muscle Mass | | Obesity | | Sarcopenia | | Sarcopenic Obesity | |
|---|---|---|---|---|---|---|---|---|---|---|
| | Present | Absent | Present | Absent | Present | Absent | Present | Absent | Present | Absent |
| | $\bar{X} \pm SD$ | $\bar{X} \pm SD$ | $\bar{X} \pm SD$ | $\bar{X} \pm SD$ | $\bar{X} \pm SD$ | $\bar{X} \pm SD$ | $\bar{X} \pm SD$ | $\bar{X} \pm SD$ | $\bar{X} \pm SD$ | $\bar{X} \pm SD$ |
| Lean mass/Fat mass | 2.1±0.61 | 2.5±1.17 | 2.6±1.2[a] | 2.5±1.1 | **1.7±0.29**[*] | **2.9±1.20**[*] | 2.2±0.72 | 2.5±1.14 | **1.5±0.24**[*] | **2.9±1.23**[*] |

[*]: unpaired Student t-test between present and absent subgroups for the same diagnostic, p≤0.05 (highlighted in bold). Cutoffs applied: for Pre-sarcopenia diagnostic, HGS<16kg for women, HGS<27kg for men [10]; for low muscle mass diagnostic, ALM<15kg for women, ALM<20kg for men [10]; for obesity diagnostic, FMI>13kg/m$^2$ for women, FMI>9kg/m$^2$ for men [14]. For sarcopenia diagnostic, presence of diagnostic was applied if there is a concomitant presence of pre-Sarcopenia and low muscle mass diagnostics, and absence of sarcopenia diagnostic in the absence of both, pre-Sarcopenia and low muscle mass diagnostics [10]. For sarcopenic obesity diagnostic, presence of diagnostic was applied if there is a concomitant presence of low muscle mass and obesity diagnostics, and absence of sarcopenic obesity diagnostic in the absence of both, low muscle mass and obesity diagnostics. Appendicular lean mass, appendicular lean mass index, fat mass, fat mass index, fat mass of the trunk, and lean mass/fat mass, by dual energy X-ray absorptiometry analysis. Phase angle and over-hydration by bioelectrical impedance analysis.

### Prospective analysis

In the prospective analysis, 87 patients were reevaluated after 10±2 months (S1 Fig). Patients with and without the official conditions (classification obtained in the first evaluation) were compared for clinical, anthropometric, and body composition changes (value of the second assessment–value of the first assessment) (S1 Table). As there were only 2 pre-sarcopenic obese patients in the prospective assessment, analysis was not done for this condition.

Both sexes tended to lose weight, HGS, and muscle mass and gain total and central adiposity with time. All patients, independent of pre-diagnosis, lost muscle mass. The ones with pre-existing low muscle mass also lost fat mass. All the others gained total and central adiposity. These results are supported by S1 Fig, in which it is possible to observe that there was a tendency for patients with normal body composition in the first evaluation to be diagnosed with a worse body composition in the second evaluation; changes to a better body composition occurred more rarely.

As only few patients participated in the second assessment, to evaluate if there was some selection bias that could influenced the direction of body composition changes, we compared cross-sectional data from patients that were evaluated only in the first assessment with the ones that participated in the first and second assessment (S2 Table). No statistical difference was found between groups.

### Diagnostic capacity of bed-side measurements

ROC curves for females and males for low muscle mass diagnosis are presented in S2 and S3 Figs, respectively, with the highest AUC for AFFM and pFFM, and among anthropometric measures, for CC. ROC curves for females and males for obesity diagnosis are shown in S4 and S5 Figs, respectively, with the highest AUC for FMIBCM and pFMI, and among anthropometric data, for body mass index, WC, and WC/H.

The cutoffs proposed in the present study for evaluation of low muscle mass (Table 3) and obesity (Table 4) were different from those proposed for the population without CKD and between men and women. AFFM, pFFM, and CC were the most efficient measures to identify low muscle mass for both sexes. As expected, these measures also showed the lowest OR for low muscle mass diagnosis (Table 2) and the highest correlation coefficients with ALM (S3 Table). FMIBCM, pFMI, and WC/H were the most efficient measures to identify obesity in both sexes. As expected, these measures also showed the highest OR for obesity diagnosis (Table 2) and the highest correlation coefficients with FMI (S3 Table).

**Table 2. ORs of the clinical, muscle, adiposity and bioelectrical data for low muscle mass[1] and obesity[2] risk.**

*Low Muscle Mass[1]*

| | Sex | OR (95%CI) | Sex | OR (95%CI) |
|---|---|---|---|---|
| Age (years) | Female | 0.99 (0.95, 1.02) | Male | 1.00 (0.97, 1.04) |
| Chronic kidney disease treatment group[3] | Female | 1.38 (0.99, 1.91) | Male | 1.26 (0.95, 1.65) |
| Dialysis or kidney transplant time (month) | Female | 1.00 (0.99, 1.01) | Male | 1.00 (0.99, 1.00) |
| Estimated glomerular filtration rate (ml/min/1.73m$^2$) | Female | 1.02 (1.00, 1.04) | Male | 1.02 (1.00, 1.03) |
| KT/V | Female | 1.25 (0.49, 3.17) | Male | 1.00 (0.29, 3.35) |
| Diabetes Mellitus[4] | Female | 0.74 (0.33, 1.66) | Male | 0.60 (0.50, 1.40) |
| Weight (kg) | Female | 0.81 (0.76, 0.87)* | Male | 0.85 (0.81, 0.90)* |
| Body mass index (kg/m$^2$) | Female | 0.73 (0.65, 0.82)* | Male | 0.72 (0.64, 0.81)* |
| Lean mass (kg) by dual energy X-ray absorptiometry analysis | Female | 0.44 (0.33, 0.60)* | Male | 0.53 (0.42, 0.66)* |
| Fat mass (kg) by dual energy X-ray absorptiometry analysis | Female | 0.85 (0.80, 0.90)* | Male | 0.90 (0.85, 0.94)* |
| Hand grip strength (kg) | Female | 0.87 (0.81, 0.94)* | Male | 0.92 (0.87, 0.96)* |
| Mid arm muscle circumference (cm) | Female | 0.67 (0.56, 0.80)* | Male | 0.70 (0.59, 0.83)* |
| Arm muscle area (cm$^2$) | Female | 0.90 (0.86, 0.94)* | Male | 0.91 (0.90, 0.95)* |
| Corrected arm muscle area (cm$^2$) | Female | 0.90 (0.86, 0.94)* | Male | 0.91 (0.88, 0.95)* |
| Adductor pollicis muscle thickness (mm) | Female | 0.83 (0.75, 0.92)* | Male | 0.76 (0.68, 0.85)* |
| Calf circumference (cm) | Female | 0.54 (0.43, 0.67)* | Male | 0.61 (0.51, 0.73)* |
| Phase angle (°) | Female | 0.65 (0.43, 0.98)* | Male | 0.87 (0.59, 1.30) |
| Body cell mass (kg) | Female | 0.70 (0.61, 0.81)* | Male | 0.80 (0.73, 0.88)* |
| Body cell mass index (kg/m$^2$) | Female | 0.53 (0.39, 0.72)* | Male | 0.70 (0.56, 0.98)* |
| Fat free mass body composition monitor (kg) | Female | 0.77 (0.70, 0.86)* | Male | 0.85 (0.79, 0.91)* |
| Fat free mass body composition monitor (%) | Female | 1.05 (1.01, 1.08)* | Male | 1.05 (1.01, 1.07)* |
| Fat free mass index body composition monitor (kg/m$^2$) | Female | 0.62 (0.49, 0.78)* | Male | 0.76 (0.65, 0.90)* |
| Predicted fat free mass (kg) | Female | 0.43 (0.32, 0.59)* | Male | 0.64 (0.55, 0.76)* |
| Predicted fat free mass (%) | Female | 1.20 (1.10, 1.31)* | Male | 1.03 (0.97, 1.09) |
| Predicted fat free mass index (kg/m$^2$) | Female | 0.43 (0.31, 0.58)* | Male | 0.46 (0.34, 0.61)* |
| Appendicular fat free mass (kg) | Female | 0.18 (0.10, 0.34)* | Male | 0.32 (0.22, 0.48)* |

*Obesity[2]*

| | Sex | OR (95%CI) | Sex | OR (95%CI) |
|---|---|---|---|---|
| Age (years) | Female | 1.11 (1.03, 1.20)* | Male | 1.03 (0.99, 1.07) |
| Chronic kidney disease treatment group[3] | Female | 0.78 (0.52, 1.20) | Male | 0.89 (0.66, 1.20) |
| Dialysis or kidney transplant time (month) | Female | 1.00 (0.99, 1.01) | Male | 1.00 (0.99, 1.01) |
| Estimated glomerular filtration rate (ml/min/1.73m$^2$) | Female | 0.99 (0.97, 1.01) | Male | 0.99 (0.97, 1.01) |
| KT/V | Female | 0.41 (0.72, 2.44) | Male | 0.98 (0.20, 4.81) |
| Diabetes mellitus[4] | Female | 4.41 (1.71, 11.38)* | Male | 2.49 (1.13, 5.48)* |
| Weight (kg) | Female | 1.15 (1.08, 1.21)* | Male | 1.14 (1.09, 1.20)* |
| Body mass index (kg/m$^2$) | Female | 2.05 (1.50, 2.78)* | Male | 2.32 (1.70, 3.16)* |
| Hand grip strength (kg) | Female | 0.92 (0.84, 1.01) | Male | 0.98 (0.94, 1.02) |
| Phase angle (°) by bioelectrical impedance analysis | Female | 0.95 (0.59, 1.54) | Male | 1.03 (0.68, 1.53) |
| Body cell mass (kg) by bioelectrical impedance analysis | Female | 0.90 (0.80, 1.02) | Male | 0.98 (0.91, 1.05) |
| Mid arm circumference (cm) | Female | 1.60 (1.32, 1.94)* | Male | 1.77 (1.45, 2.15)* |
| Waist circumference (cm) | Female | 1.23 (1.13, 1.33)* | Male | 1.30 (1.18, 1.44)* |
| Waist circumference/Height | Female | 8.87 (2.24, 3.51)* | Male | 2.17 (1.68, 2.80)* |
| Triceps skin fold thickness (mm) | Female | 1.18 (1.11, 1.25)* | Male | 1.40 (1.24, 1.57)* |
| Fat mass body composition monitor (kg) | Female | 1.59 (1.29, 1.96)* | Male | 1.36 (1.22, 1.51)* |
| Fat mass body composition monitor (%) | Female | 1.50 (1.27, 1.76)* | Male | 1.35 (1.21, 1.51)* |
| Fat mass index body composition monitor (kg/m$^2$) | Female | 4.58 (2.13, 9.86)* | Male | 3.15 (2.10, 4.78)* |
| Predicted fat mass (kg) | Female | 1.51 (1.27, 1.78)* | Male | 1.56 (1.32, 1.83)* |

*(Continued)*

**Table 2.** (Continued)

*Low Muscle Mass*[1]

|  | Sex | OR (95%CI) | Sex | OR (95%CI) |
|---|---|---|---|---|
| Predicted fat mass (%) | Female | 2.37 (1.64, 3.41)* | Male | 1.73 (1.43, 2.10)* |
| Predicted fat mass index (kg/m$^2$) | Female | 6.43 (2.41, 17.11)* | Male | 5.48 (2.86, 10.52)* |

[1]ALM with dual energy X-ray absorptiometry measurement, for women, ALM<15kg, for men, ALM<20kg [10].

[2]FMI with dual energy X-ray absorptiometry measurement, for women, FMI>13kg/m$^2$, for men, FMI>9kg/m$^2$ [14].

[3]NDD CKD group as reference.

[4]Presence of diagnostic of Diabetes Mellitus as reference.

*p≤0.05. Phase angle, body cell mass, body cell mass index, fat free mass body composition monitor, fat free mass index body composition monitor, predicted fat free mass, predicted fat free mass index, appendicular fat free mass, fat mass body composition monitor, fat mass index body composition monitor, predicted fat mass and predicted fat mass index, by bioelectrical impedance. Appendicular fat free mass by Sergi equation [20]; predicted fat free mass and predicted fat mass by Bellafronte equation [22]; Fat free mass body composition monitor and fat mass body composition monitor by bioelectrical impedance from body composition monitor (Fresenius Medical Care).

For all conditions, BIS data had a better performance than anthropometric variables.

Data about sensitivity and specificity of the best bed-side measurements for low muscle mass (AFFM and CC), obesity (FMIBCM and WC/H), sarcopenia (AFFM+HGS and CC +HGS) and sarcopenic obesity (FMIBCM+AFFM) in each CKD subgroups are presented in S4 Table.

The bed-side variables with the best performance for low muscle mass and obesity were chosen for sensitivity and specificity analysis of sarcopenia and sarcopenic obesity (Table 5). AFFM+HGS, pFFM+HGS, and CC+HGS presented almost the same sensitivity and specificity for sarcopenia diagnosis. FMIBCM+AFFM and FMIBCM+pFFM followed by pFMI+AFFM and pFMI+pFFM, presented the best sensitivity and specificity for sarcopenic obesity diagnosis. The analysis stratified by CKD group showed wide 95%CI for diagnosis of sarcopenia and sarcopenic obesity (S4 Table).

Agreement between diagnoses based on bed-side measure and official ones varied from moderate to almost perfect (Table 6).

## Discussion

This study's main results were the proposed cutoffs for muscle and adiposity bed-side measures/indexes for low muscle mass, sarcopenia, obesity, and sarcopenic obesity diagnosis in CKD patients. We found different cutoff values from those proposed for the general population. This is the first study, to our knowledge, that suggests anthropometric and bioelectrical impedance measures as tools for diagnosis of the four conditions previously cited in patients with CKD under different treatment modalities and applying the new definition of sarcopenia. Thus, we believe that our work is of great relevance in clinical practice. In addition, the methodology used was rigorous, and included DXA analysis in all patients and a prospective assessment. We included CKD patients in NDD, HD, PD, and KTx therapy, allowing participation of subjects with different degrees of adiposity and over-hydration, known to be interfering factors in the accuracy of anthropometric and bioelectrical impedance measurements [31, 32].

Of all the measures evaluated, AFFM and CC presented the best performance for low muscle mass and sarcopenia (in addition to HGS) diagnosis. FMIBCM and WC/H showed the best performance for obesity diagnosis and, FMIBCM+AFFM, for sarcopenic obesity diagnosis. Analysis with stratification of CKD subgroups showed good performance of AFFM, CC, FMIBCM and WC/H for all CKD groups, mainly for BIS measurements. On the other hand,

**Table 3. Area under the ROC curve, cutoff, sensitivity, and specificity of the muscle measures/indexes to identify low muscle mass[1].**

| Variables | Sex | Area under the ROC curve (95%IC) | Cutoff | Sensitivity, % (95%CI) | Specificity, % (95%CI) |
|---|---|---|---|---|---|
| MAMC (cm) | Female | 0.78 (0.70, 0.86)* | ≤22.03 | 65.4 (54.0, 75.7) | 89.4 (76.9, 96.5) |
| | Male | 0.76 (0.67, 0.84)* | ≤26.92 | 80.0 (66.3, 90.0) | 61.2 (50.0, 71.6) |
| AMA (cm$^2$) | Female | 0.78 (0.70, 0.87)* | ≤38.65 | 65.4 (54.0, 75.7) | 89.4 (76.9, 96.5) |
| | Male | 0.76 (0.67, 0.84)* | ≤57.68 | 80.0 (66.3, 90.0) | 61.2 (50.0, 71.6) |
| cAMA (cm$^2$) | Female | 0.78 (0.70, 0.87)* | ≤32.15 | 65.4 (54.0, 75.7) | 89.4 (76.9, 96.5) |
| | Male | 0.76 (0.67, 0.84)* | ≤47.68 | 80.0 (66.3, 90.0) | 61.2 (50.0, 71.6) |
| APMT (mm) | Female | 0.72 (0.63, 0.81)* | ≤18.33 | 72.5 (61.4, 81.9) | 65.2 (49.8, 78.6) |
| | Male | 0.81 (0.74, 0.88)* | ≤24.33 | 89.8 (77.8, 96.6) | 60.5 (49.3, 70.8) |
| CC (cm) | Female | 0.89 (0.84, 0.95)* | ≤35.5 | 76.5 (65.8, 85.2) | 93.6 (82.5, 98.7) |
| | Male | 0.85 (0.79, 0.91)* | ≤37.0 | 88.0 (75.7, 95.5) | 69.4 (58.5, 79.0) |
| PhA (°) | Female | 0.60 (0.50, 0.70)* | ≤5.43 | 46.9 (35.7, 58.3) | 74.5 (59.7, 86.1) |
| | Male | 0.55 (0.45, 0.65)* | ≤6.32 | 56.0 (41.3, 70.0) | 59.3 (48.2, 69.8) |
| BCM (kg) | Female | 0.78 (0.70, 0.86)* | ≤17.6 | 75.3 (64.5, 84.2) | 66.0 (50.7, 79.1) |
| | Male | 0.76 (0.68, 0.84)* | ≤29.2 | 90.0 (78.2, 96.7) | 51.2 (40.1, 62.1) |
| BCMI (kg/m$^2$) | Female | 0.73 (0.64, 0.82)* | ≤7.2 | 70.4 (59.2, 80.0) | 68.1 (52.9, 80.9) |
| | Male | 0.65 (0.56, 0.74)* | ≤8.67 | 54.0 (39.3, 68.2) | 74.4 (63.9, 83.2) |
| FFMBCM (kg) | Female | 0.79 (0.71, 0.87)* | ≤31.7 | 70.4 (59.2, 80.0) | 72.3 (57.4, 84.4) |
| | Male | 0.77 (0.69, 0.85)* | ≤49.9 | 94.0 (83.5, 98.7) | 51.2 (40.1, 62.1) |
| FFMBCM (%) | Female | 0.36 (0.27, 0.46) | - | - | - |
| | Male | 0.34 (0.24, 43) | - | - | - |
| FFMIBCM (kg/m$^2$) | Female | 0.73 (0.65, 0.82)* | ≤13.29 | 75.3 (64.5, 84.2) | 63.8 (48.5, 77.3) |
| | Male | 0.66 (0.56, 0.75)* | ≤15.24 | 54.0 (39.3, 68.2) | 74.4 (63.9, 83.2) |
| pFFM (kg) | Female | 0.96 (0.93, 0.99)* | ≤35.02 | 87.6 (78.5, 93.9) | 93.6 (82.5, 98.7) |
| | Male | 0.94 (0.89, 0.98)* | ≤46.42 | 95.9 (86.0, 99.5) | 85.9 (76.6, 92.5) |
| pFFM (%) | Female | 0.26 (0.18, 0.35) | - | - | - |
| | Male | 0.42 (0.32, 0.52) | - | - | - |
| pFFMI (kg/m$^2$) | Female | 0.83 (0.75, 0.90)* | ≤13.68 | 64.2 (52.8, 74.6) | 89.4 (76.9, 96.5) |
| | Male | 0.86 (0.79, 0.92)* | ≤16.85 | 87.8 (75.2, 95.4) | 76.5 (66.0, 85.0) |
| AFFM (kg) | Female | 0.96 (0.94, 0.99)* | ≤15.87 | 90.1 (81.5, 95.6) | 95.7 (85.5, 99.5) |
| | Male | 0.94 (0.89, 0.98)* | ≤21.43 | 98.0 (89.4, 99.9) | 83.7 (74.2, 90.8) |

[1]ALM by Dual energy X-ray absorptiometry analysis, with ALM<15kg for women and ALM <20kg for men [10].

*p ≤ 0.05. FFMBCM (%) and pFFM (%): AUC < 0.50 (it is not possible to define a cutoff). AFFM, appendicular fat free mass; ALM, appendicular lean mass; AMA, arm muscle area; APTM, adductor pollicis muscle thickness; BCM, body cell mass; BCMI, body cell mass index; cAMA, corrected arm muscle area; CC, calf circumference; FFMBCM, fat free mass by body composition monitor; FFMIBCM, fat free mass index by body composition monitor; MAMC, mid-arm muscle circumference; pFFM, predicted fat free mass; pFFMI, predicted fat free mass index; PhA, phase angle; ROC, receiver operating characteristic. PhA, BCM, BCMI, FFMBCM, FFMIBCM, pFFM, pFFMI and AFFM measures by bioelectrical impedance. AFFM by Sergi equation [20] and pFFM by Bellafronte equation [22]. FMBCM by bioelectrical impedance from body composition monitor (Fresenius Medical Care).

although a good sensibility and specificity, the wide 95%CI of sarcopenia and sarcopenic obesity diagnosis by bed-side measurements in each CKD group show a need of validation in a larger sample size before routine use of these measurements.

AFFM was calculated by the Sergi equation [20], recommended by the European Working Group on Sarcopenia in Older People [21] for prediction of AFFM by bioelectrical impedance, as estimates of muscle mass differ when different devices, brands, and reference populations are used [31, 32]. The Sergi equation also showed the best performance among five bioelectrical impedance equations applied in adults from Australia [33].

**Table 4. Area under the ROC curve, cutoff, sensitivity, and specificity of the adiposity measures/indexes to identify obesity[1].**

| Variables | Sex | Area under the ROC curve (95%IC) | Cutoff | Sensitivity, % (95%CI) | Specificity, % (95%CI) |
|---|---|---|---|---|---|
| Weight (kg) | Female | 0.89 (0.84, 0.95)* | >69.4 | 91.3 (72.0, 98.9) | 78.1 (69.0, 85.6) |
| | Male | 0.87 (0.82, 0.93)* | >72.7 | 100.0 (91.4, 100.0) | 60.0 (49.4, 69.9) |
| BMI (kg/m²) | Female | 0.96 (0.93, 0.99)* | >29.1 | 90.7 (73.1, 94.9) | 86.7 (76.5, 91.6) |
| | Male | 0.96 (0.93, 0.98)* | >28.87 | 87.2 (73.9, 94.3) | 87.6 (80.4, 92.0) |
| MAC (cm) | Female | 0.92 (0.87, 0.97)* | >33.2 | 86.9 (66.4, 97.2) | 81.9 (73.2, 88.7) |
| | Male | 0.90 (0.84, 0.95)* | >33.0 | 85.4 (70.8, 94.4) | 86.2 (77.5, 92.4) |
| WC (cm) | Female | 0.92 (0.87, 0.97)* | >103.2 | 91.3 (72.0, 98.9) | 81.9 (73.2, 88.7) |
| | Male | 0.93 (0.89, 0.97)* | >103.5 | 95.1 (83.5, 99.4) | 78.5 (66.8, 86.3) |
| WC/H | Female | 0.94 (0.90, 0.98)* | >0.66 | 91.3 (72.0, 98.9) | 83.8 (75.3, 90.3) |
| | Male | 0.95 (0.92, 0.98)* | >0.60 | 95.1 (83.5, 99.4) | 79.6 (69.9, 87.2) |
| TSF (mm) | Female | 0.90 (0.84, 0.96)* | >31.33 | 86.4 (65.1, 97.1) | 82.7 (74.0, 89.4) |
| | Male | 0.92 (0.88, 0.97)* | >15.33 | 95.1 (83.5, 99.4) | 77.7 (67.9, 85.3) |
| ABSI | Female | 0.34 (0.23, 0.45) | - | - | - |
| | Male | 0.47 (0.37, 0.57) | - | - | - |
| Conicity index | Female | 0.66 (0.55, 0.77)* | >1.39 | 78.3 (56.3, 92.5) | 52.4 (42.4, 62.2) |
| | Male | 0.71 (0.63, 0.80)* | >1.32 | 97.6 (87.1, 99.9) | 38.7 (28.8, 49.4) |
| FMBCM (kg) | Female | 0.97 (0.94, 0.94)* | >30.2 | 100.0 (85.2, 100.0) | 86.7 (78.6, 92.5) |
| | Male | 0.93 (0.89, 0.97)* | >25.8 | 87.8 (73.8, 95.9) | 87.4 (79.0, 93.3) |
| FMBCM (%) | Female | 0.94 (0.89, 0.99)* | >43.29 | 91.3 (72.0, 98.9) | 94.3 (88.0, 97.9) |
| | Male | 0.91 (0.86, 0.96)* | >30.78 | 87.8 (73.8, 95.9) | 81.1 (71.7, 88.4) |
| FMIBCM (kg/m²) | Female | 0.99 (0.98, 1.00)* | >12.58 | 100.0 (85.2, 100.0) | 93.3 (86.7, 97.3) |
| | Male | 0.96 (0.93, 0.99)* | >8.82 | 92.7 (80.1, 98.5) | 88.4 (80.2, 94.1) |
| pFM (kg) | Female | 0.95 (0.92, 0.99)* | >28.58 | 95.6 (78.1, 99.9) | 83.8 (75.3, 90.3) |
| | Male | 0.95 (0.91, 0.98)* | >24.91 | 92.7 (80.1, 98.5) | 87.1 (78.5, 93.2) |
| pFM (%) | Female | 0.96 (0.90, 1.00)* | >41.05 | 91.3 (72.0, 98.9) | 93.3 (86.7, 97.3) |
| | Male | 0.93 (0.89, 0.98)* | >29.97 | 85.4 (70.8, 94.4) | 87.1 (78.5, 93.2) |
| pFMI (kg/m²) | Female | 0.98 (0.97, 1.00)* | >12.2 | 95.6 (78.1, 99.9) | 92.4 (85.5, 96.7) |
| | Male | 0.97 (0.95, 0.99)* | >8.76 | 92.7 (80.1, 98.5) | 93.5 (86.5, 97.6) |

[1]FMI by Dual energy X-ray absorptiometry analysis, with FMI>13kg/m² for women and FMI>9kg/m² for men [14].

*p ≤ 0.05. ABSI: AUC < 0.50 (it is not possible to define a cutoff). ABSI, a body adiposity index; BMI, body mass index; FMBCM, fat mass body composition monitor; FMIBCM, fat mass index body composition monitor; MAC, mid-arm circumference; pFM, predicted fat mass; pFMI, predicted fat mass index; ROC, receiver operating characteristic; WC, waist circumference; WC/H, waist circumference for height ratio. FMBCM, FMIBCM, pFM and pFMI measures by bioelectrical impedance. pFM by Bellafronte equation [22] and FMBCM by bioelectrical impedance from body composition monitor (Fresenius Medical Care).

The other BIS measures with good performances were pFFM and pFMI, calculated with equations developed by our group [22] in CKD patients with DXA FFM and FM data as reference. Our pFFM and pFM showed lower limits of agreement than FFMBCM and FMBCM, and a better performance with an increase from 30 to 55% for FFM, and from 39 to 63% for FM in the percentage of residuals lower than 2 kg (DXA–BIS data).

FFMI by bioelectrical impedance analysis predicted a lower risk of death or cardiovascular events among NDD treatment CKD patients [9], adding even more value in BIS measurements for clinical purposes.

Phase angle has been significantly associated with death in NDD treatment CKD patients [34]. Despite recognized clinical applicability [35], phase angle did not present good capacity to diagnose low muscle mass and sarcopenia.

**Table 5. Sensitivity and specificity of bed-side measures/indexes to identify sarcopenia[1] and sarcopenic obesity[2].**

| Variables | Sex | Cutoff | Sensitivity, % (95%CI) | Specificity, % (95%CI) |
|---|---|---|---|---|
| *Sarcopenia* | | | | |
| AFFM+HGS | *Female* | AFFM≤15.87 + HGS<16 | 84.6 (78.4; 90.9) | 99.1 (97.5; 100.0) |
| | *Male* | AFFM≤21.43 + HGS<27 | 1.00 (1.00; 1.00) | 99.2 (97.7; 100.0) |
| pFFM+HGS | *Female* | pFFM≤35.02 + HGS<16 | 84.6 (78.4; 90.9) | 99.1 (97.5; 100.0) |
| | *Male* | pFFM≤46.42 + HGS<27 | 1.00 (1.00; 1.00) | 1.00 (1.00; 1.00) |
| CC+HGS | *Female* | CC≤35.5 + HGS<16 | 84.6 (78.4; 90.9) | 99.1 (97.5; 100.0) |
| | *Male* | CC≤37.0 + HGS<27 | 1.00 (1.00; 1.00) | 1.00 (1.00; 1.00) |
| *Sarcopenic Obesity* | | | | |
| FMIBCM+AFFM | *Female* | FMIBCM>12.58 + AFFM≤15.87 | 75.0 (67.5; 82.5) | 97.5 (94.8; 100.0) |
| | *Male* | FMIBCM>8.82 + AFFM≤21.43 | 75.0 (67.7; 82.3) | 95.3 (91.8; 98.9) |
| FMIBCM+pFFM | *Female* | FMIBCM>12.58 + pFFM≤35.02 | 75.0 (67.5; 82.5) | 97.5 (94.8; 100.0) |
| | *Male* | FMIBCM>8.82 + pFFM≤46.42 | 75.0 (67.7; 82.3) | 96.8 (93.9; 99.8) |
| FMIBCM+CC | *Female* | FMIBCM>12.58 + CC≤35.5 | 12.5 (6.8; 18.2) | 98.3 (96.1; 100.0) |
| | *Male* | FMIBCM>8.82 + CC≤37.0 | 50.0 (41.6; 58.4) | 96.1 (92.8; 99.3) |
| pFMI+AFFM | *Female* | pFMI>12.2 + AFFM≤15.87 | 62.5 (54.1; 70.8) | 98.3 (96.1; 100.0) |
| | *Male* | pFMI>8.76 + AFFM≤21.43 | 62.5 (54.3; 70.7) | 96.0 (92.7; 99.3) |
| pFMI+pFFM | *Female* | pFMI>12.2 + pFFM≤35.02 | 62.5 (54.1; 70.8) | 98.3 (96.1; 100.0) |
| | *Male* | pFMI>8.76 + pFFM≤46.42 | 62.5 (54.3; 70.7) | 96.8 (93.9; 99.8) |
| pFMI+CC | *Female* | pFMI>12.2 + CC≤35.5 | 12.5 (6.8; 18.2) | 98.3 (96.1; 100.0) |
| | *Male* | pFMI>8.76 + CC≤37.0 | 50.0 (41.5; 58.5) | 96.8 (93.9; 99.8) |
| WC/H+AFFM | *Female* | WC/H>0.66 + AFFM≤15.87 | 50.0 (41.4; 58.6) | 95.9 (92.4; 99.3) |
| | *Male* | WC/H>0.60 + AFFM≤21.43 | 75.0 (67.7; 82.3) | 92.8 (85.0; 97.2) |
| WC/H+pFFM | *Female* | WC/H>0.66 + pFFM≤35.02 | 50.0 (41.4; 58.6) | 95.9 (94.4; 99.3) |
| | *Male* | WC/H>0.60 + pFFM≤46.42 | 75.0 (67.7; 82.3) | 93.6 (89.5; 97.8) |
| WC/H+CC | *Female* | WC/H>0.66 + CC≤35.5 | 12.5 (6.8; 18.2) | 95.9 (92.4; 99.3) |
| | *Male* | WC/H>0.60 + CC≤37.0 | 50.0 (41.5; 58.5) | 92.9 (88.5; 97.2) |

[1]ALM by Dual energy X-ray absorptiometry analysis, with ALM<15kg for women and ALM <20kg for men [10].

[2]FMI by Dual energy X-ray absorptiometry analysis, with FMI>13kg/m$^2$ for women and FMI>9kg/m$^2$ for men [14]. AFFM, appendicular fat free mass; CC, calf circumference; FMIBCM, fat mass index body composition monitor; pFFM, predicted fat free mass; pFMI, predicted fat mass index; WC/H, waist circumference for height ratio. AFFM, FMIBCM, pFMI and pFFM measures by bioelectrical impedance. AFFM by Sergi equation [20], pFM and pFFM by Bellafronte equation [22] and FMBCM by bioelectrical impedance from body composition monitor (Fresenius Medical Care).

Carnavale and collaborators (2018) [36] found that mid arm muscle circumference (women <18.6 cm and men <22.3 cm) had a good performance predicting low muscle mass in older people. However, among the anthropometric measurements of this study, CC had the best performance. For elderly Japanese [37], CC less than 34 cm for men and 33 cm for women was associated with low muscle mass. Our cutoffs for mid arm muscle circumference and CC differed from both studies, showing that cutoffs are population-specific. Adding more importance to the measurement, CC was shown to predict worse clinical outcomes in HD patients [38].

For obesity diagnosis, WC/H showed a slightly better performance than WC. Body mass index also presented good AUC but its inability to distinguish muscle mass from fat mass is well-known, and a higher OR for obesity was found for WC/H. Also, as high WC presents more consistent results with higher mortality rates [39] and poor physical function [40] in CKD patients, we think this measurement could add more clinical significance.

The female subgroup in our study was at increased nutritional risk, with twice the prevalence of low muscle mass and sarcopenia than men. The higher participation of women in the

**Table 6. Analysis of agreement between official and bed-side diagnostics of low muscle mass, sarcopenia, obesity and sarcopenic obesity.**

| Low Muscle Mass | | | Sarcopenia | | | Obesity | | | Sarcopenic Obesity | | |
|---|---|---|---|---|---|---|---|---|---|---|---|
| Bed-side diagnostic | Female | Male | Bed-side diagnostic | Female | Male | Bed-side diagnostic | Female | Male | Bed-side diagnostic | Female | Male |
| Low pFFM | 0.76* | 0.77* | Low pFFM + Low HGS | 0.81* | 0.81* | High pFMI | 0.77* | 0.83* | High pFMI + Low pFFM | 0.69* | 0.78* |
| Low AFFM | 0.82* | 0.76* | Low AFFM + Low HGS | 0.86* | 0.80* | High FMIBCM | 0.83* | 0.77* | High pFMI + Low AFFM | 0.74* | 0.75* |
| Low CC | 0.64* | 0.53* | Low CC + Low HGS | 0.71* | 0.59* | High WC/H | 0.60* | 0.67* | High FMIBCM+ Low pFFM | 0.73* | 0.73* |
| | | | | | | | | | High FMIBCM+ Low AFFM | 0.78* | 0.71* |

*Kappa coefficient of agreement between official and bed-side diagnostic, p≤0.05. Official low muscle mass diagnostic: for women, ALM<15kg; for men, ALM<20kg [10]. Official sarcopenia diagnostic: for women, ALM<15kg and HGS<16kg; for men, ALM<20kg and HGS<27kg [10]. Official obesity diagnostic: for women, FMI>13kg/m$^2$; for men, FMI>9kg/m$^2$[14]. Official sarcopenic obesity: for women, ALM<15kg and FMI>13kg/m$^2$; for men, ALM<20kg and FMI>9kg/m$^2$. For official diagnostics, body composition measures were performed with dual energy X-ray absorptiometry. Bed-side low muscle mass diagnostic: for women, pFFM≤35.02kg, AFFM≤15.87kg and CC≤35.5cm; for men, pFFM≤46.42kg, AFFM≤21.43kg and CC≤37cm. Bed-side sarcopenic diagnostic: for women, pFFM≤35.02kg and HGS<16kg, AFFM≤15.87kg and HGS<16kg, CC≤35.5cm and HGS<16kg; for men, pFFM≤46.42kg and HGS<20kg, AFFM≤21.43kg and HGS<20kg, CC≤37cm and HGS<20kg. Bed-side obesity diagnostic: for women, pFMI>12.20kg/m$^2$, FMIBCM>12.58kg/m$^2$ and WC/H>0.66; for men, pFMI>8.76kg/m$^2$, FMIBCM>8.82kg/m$^2$ and WC/H>0.60. Bed-side sarcopenic obesity: for women, pFMI>12.20kg/m$^2$ and AFFM≤15.87kg, FMIBCM>12.58kg/m$^2$ and pFFM≤35.02kg, FMIBCM>12.58kg/m$^2$ and AFFM≤15.87kg; for men, pFMI>8.76kg/m$^2$ and pFFM≤46.42kg, pFMI>8.76kg/m$^2$ and AFFM≤21.43kg, FMIBCM>8.82kg/m$^2$ and pFFM≤46.42kg, FMIBCM>8.82kg/m$^2$ and AFFM≤21.43kg. AFFM, appendicular fat free mass; ALM, appendicular lean mass; CC, calf circumference; FMIBCM, fat mass index body composition monitor; HGS, hand grip strength; pFFM, predicted fat free mass; pFMI, predicted fat mass index. pFFM, AFFM, pFMI and FMIBCM data by bioelectrical impedance analysis. AFFM by Sergi equation [20]. pFFM and pFMI by Bellafronte equation [22]. FMIBCM by bioelectrical impedance from body composition monitor (Fresenius Medical Care).

HD and PD groups, which have a more compromised nutritional status, may have influenced the results. Zhou and collaborators (2018) [41] reported that more comorbidities, age, and female sex were associated with less lean mass and more fat mass, although they found men to be more affected by sarcopenia than women.

Our dialysis patients, especially those in HD, had the highest muscle depletion and prevalence of low muscle mass and sarcopenia. As a long dialysis vintage is related to high number of protein energy wasting categories [42], the longer HD therapy, compared to the PD group, may have contributed to the worst nutritional status. In addition, the loss of amino acids and proteins in the dialysate in HD, with a higher inflammatory and metabolic acidosis condition, increase catabolism and decrease protein synthesis [43]. PD patients were also younger, and age is recognized as a factor positively associated with decreased muscle mass [44]. Other studies also found that HD patients had the greatest muscle loss [11]. Furthermore, loss of lean body mass is associated with a decrease in glomerular filtration rate [41]. Our findings were in accordance with those ones, as our NDD patients also presented high prevalence of low muscle mass, but in a lower degree than the dialysis group.

A high prevalence of low muscle mass was found in KTx patients, similar to that of the PD group, demonstrating the maintenance of a compromised nutritional status even after a long post-transplant time. Our data were in agreement with previous findings that low muscle mass, function, and physical performance are common conditions among KTx patients [8].

Beyond body composition analysis, some studies have reported a relationship of body mass index and nutritional status changes over time with higher mortality rates [45, 46]. HD patients with declining body mass index have higher rates of mortality compared to body mass index stable patients [46]. Patients with stable weight had a higher survival, but weight maintenance does not mean body composition maintenance. As shown by our data, CKD patients tended to lose muscle mass and gain fat mass independently of sex and presence or absence of a condition, signaling a deterioration of nutritional status, which was, as also as persistent poor nutrition status, associated with poor quality of life and higher mortality [45].

Almost one third of NDD patients and most of the KTx group was obese, in agreement with the known continuous increase in obesity prevalence in end-stage CKD [10]. Patients in CKD stages 3b and 4 were found to have higher FMI than patients in CKD stage 5 [11], although we did not find differences in FMI among CKD groups.

In our sample, men, NDD, and KTx patients had higher prevalence of obesity and lower prevalence of low muscle mass and sarcopenia. In addition, adiposity was a protective factor for low muscle mass diagnosis, as in the obesity paradox in CKD, which is not well understood [6, 9, 12].

A previous study showed that LMI was a better risk prediction parameter of clinical outcomes than body mass index in NDD CKD patients [9], with a high lean/fat tissue ratio phenotype associated with better outcomes. On the other hand, sarcopenia was associated with increased mortality regardless of estimated glomerular filtration rate, but excess adiposity modified this association only among persons with CKD [6]. Also, insulin derangements were shown to modify the relationship between adiposity and mortality, as body fat showed a protective effect on survival only in HD patients with insulin resistance [12]. The studies evidence that interfering factors could play a role in the obesity paradox, but it is yet unclear whether increased muscle mass or increased body fat confers the survival advantage.

In turn, not only total fat but also body fat distribution appears to be important for mortality. Visceral fat is more strongly related to complications of obesity, such as proatherogenic lipid profile, than subcutaneous fat [47]. In elderly men [48] and in PD patients [39], high WC and its increase over time were also predictors of mortality. In our sample, WC and especially WC/H had the highest OR for obesity. These results confirm the great importance of central obesity.

## Conclusions

Our study demonstrated that nutritional status of CKD patients is usually compromised, primarily in dialysis CKD but also in NDD and KTx patients. Also, renal patients tended to worse body composition with time. Given the well-known strong association of poor nutritional status with a worse clinical prognosis and higher mortality, nutritional assessment has to be done frequently and as part of the regular CKD patient care. In order to make an early identification allowing interventions and improving clinical outcomes, tools with diagnostic capacity that are easily available are needed. We revealed that BIS and anthropometric measurements, such as AFFM and CC, could be used with the proposed cutoffs for low muscle mass diagnosis, and with the combination of HGS, for sarcopenia diagnosis. We also specified cutoffs for FMIBCM and WC/H for obesity and FMIBCM+AFFM for sarcopenic obesity diagnosis. Training of multiprofessional teams, however, is mandatory to carry out these simple measures that can be used routinely even in institutions that do not have high-cost equipment.

Cross-validation in a large sample size with participation of four CKD treatment groups is still needed.

## Strengths

We evaluated not only one but four conditions related to impairment of nutritional status, as low muscle mass, sarcopenia, obesity and sarcopenic obesity. Also, we have a large sample of CKD patients under the four type of treatment, as NDD, HD, PD and KTx. In addition, we applied the new definition of sarcopenia by EWGSOP with DXA analysis, a reference method for body composition evaluation, in all patients. We conducted a cross sectional and prospective study with important data of body composition trajectory in CKD. Thorough statistics were carried out allowing a comprehensive and in-depth analysis of our results. Finally, we

suggest easily available anthropometric and bioelectrical impedance measures as tools for diagnosis of the conditions previously cited.

## Limitations

Sex and age are well known important factors in determine skeletal muscle mass and body adiposity. In addition, physical function and eGFR declines with age. So, our mayor study limitation is the exclusion of patients over 60 years of age. As we evaluated only adult CKD patients, our results cannot be applied to the elderly. Validation of the measures and cut-off points in elderly are needed not only because of differences in body composition but also in the accuracy of instruments. Also, longer prospective evaluations could provide more information about the body composition trajectory of CKD patients and associations between outcomes and mortality. The PD group was underrepresented, which may have affected CKD subgroup analysis. Moreover, we applied a definition of sarcopenic obesity that takes into account only low muscle mass and obesity, not including muscle function, such as low HGS. Although there is not a consensus on sarcopenic obesity definition, which varies considerably, we think this information should be considered. Finally, the sample included only 2 patients with pre-sarcopenic obesity in the prospective assessment, which limited our analysis of body composition changes in this group.

## Supporting information

**S1 Fig. Follow up graphic from first to second assessment of patients with participation in both assessment.**
(DOCX)

**S2 Fig. Receiver operation characteristic curve for low muscle mass diagnostic in female sample.**
(DOCX)

**S3 Fig. Receiver operation characteristic curve for low muscle mass diagnostic in male sample.**
(DOCX)

**S4 Fig. Receiver operation characteristic curve for obesity diagnostic in female sample.**
(DOCX)

**S5 Fig. Receiver operation characteristic curve for obesity diagnostic in male sample.**
(DOCX)

**S1 Table. Clinical, anthropometric and body composition changes (second–first assessment) stratified by sex and official diagnostic.**
(DOCX)

**S2 Table. Clinical, anthropometric and body composition data from cross-sectional assessment of CKD patients stratified according to participation in the second assessment.**
(DOCX)

**S3 Table. Correlation between dual energy X-ray absorptiometry and bed-side measurements of muscle mass and adiposity.**
(DOCX)

**S4 Table. Sensitivity and specificity of bed-side measures/indexes to identify low muscle mass, obesity, sarcopenia and sarcopenic obesity in CKD subgroups.**
(DOCX)

## Acknowledgments

We sincerely appreciate all participants for their kind cooperation in our study. We also wish to acknowledge all of the investigators in our study and the hospital staff members.

## Author Contributions

**Conceptualization:** Natália Tomborelli Bellafronte, Guillermina Barril Cuadrado.

**Data curation:** Natália Tomborelli Bellafronte, Gabriel Ruiz Sizoto.

**Formal analysis:** Natália Tomborelli Bellafronte.

**Funding acquisition:** Paula Garcia Chiarello.

**Investigation:** Natália Tomborelli Bellafronte.

**Methodology:** Natália Tomborelli Bellafronte, Lorena Vega-Piris.

**Project administration:** Natália Tomborelli Bellafronte, Paula Garcia Chiarello, Guillermina Barril Cuadrado.

**Software:** Lorena Vega-Piris.

**Supervision:** Paula Garcia Chiarello, Guillermina Barril Cuadrado.

**Validation:** Guillermina Barril Cuadrado.

**Visualization:** Natália Tomborelli Bellafronte, Guillermina Barril Cuadrado.

**Writing – original draft:** Natália Tomborelli Bellafronte.

**Writing – review & editing:** Natália Tomborelli Bellafronte, Lorena Vega-Piris, Paula Garcia Chiarello, Guillermina Barril Cuadrado.

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
