## [Decision Letter · Decision Letter 0]

13 Oct 2020

PONE-D-20-28011

Bed-side measures for diagnosis of low muscle mass, sarcopenia, obesity, and sarcopenic obesity in patients with chronic kidney disease under non-dialysis-dependent, dialysis dependent and kidney transplant therapy

PLOS ONE

Dear Dr. Bellafronte,

Thank you for submitting your manuscript to PLOS ONE. After careful consideration, we feel that it has merit but does not fully meet PLOS ONE’s publication criteria as it currently stands. Therefore, we invite you to submit a revised version of the manuscript that addresses the points raised during the review process.

We look forward to receiving your revised manuscript.

Kind regards,

Mauro Lombardo

Academic Editor

PLOS ONE

Journal Requirements:

2. In your Methods section, please provide additional information about the participant recruitment method and the demographic details of your participants. Please ensure you have provided sufficient details to replicate the analyses such as:

a) the recruitment date range (month and year),

b) any sample size calculations,

c) a description of how participants were recruited, and d) descriptions of where participants were recruited and where the research took place.

5. Please ensure that you include a title page within your main document. You should list all authors and all affiliations as per our author instructions and clearly indicate the corresponding author.

6. Please amend your manuscript to include your abstract after the title page.

Reviewers' comments:

Reviewer's Responses to Questions

**Comments to the Author**

1. Is the manuscript technically sound, and do the data support the conclusions?

Reviewer #1: Partly

Reviewer #2: Yes

2. Has the statistical analysis been performed appropriately and rigorously? 

Reviewer #1: No

Reviewer #2: Yes

3. Have the authors made all data underlying the findings in their manuscript fully available?

Reviewer #1: Yes

Reviewer #2: Yes

4. Is the manuscript presented in an intelligible fashion and written in standard English?

Reviewer #1: Yes

Reviewer #2: Yes

5. Review Comments to the Author

Reviewer #1: This is a cross-sectional study (n 265), followed by a prospective follow-up study with 33% (n 87) of the patients initially included. The study provides a detailed description of the prevalence of obesity, sarcopenia, and sarcopenic obesity in a cohort of patients with CKD (CKD Stages 1 to 3a, PD, hemodialysis, and kidney transplantation). The paper provides relevant information comparing bed-sides measures with dual-energy X-ray absorptiometry in a cross-sectional study in CKD patients. It also tries to establish cut-off values for CKD patients. However, the diagnostic capabilities of the measurements of muscle and fat mass are analyzed mainly according to sex and without considering the different situations of renal function; this makes it difficult to assess the sensitivity and specificity of the measures in different "renal situations" and also questions the validity of the cut-off points. An analysis of the diagnostic capacity of the tests in relation to the renal situation is necessary.

In the prospective study (n 87), the renal situation of the patients is not clear. A graph comparing the initial situation and follow-up according to the initial classification (renal and sarcopenic-obesity) would also be interesting.

The study excluded patients over 60 years of age, a group that constitutes an important and growing population with a high prevalence of CKD, sarcopenia, and obesity. This point should be discussed more extensively in the discussion.

Minor review:

Abstract: the acronyms DXA and HGS must be defined in the abstract on its first use.

It would be interesting to include a table of the different measures according to the renal situation.

Page 6, lines 6 and 7, the sentence is understood, but it isn't very clear.

Reviewer #2: I was honored to review the manuscript entitled “Bed-side measures for diagnosis of low muscle mass, sarcopenia, obesity, and sarcopenic obesity in patients with chronic kidney disease under non-dialysis dependent, dialysis dependent and kidney transplant therapy” submitted to Plos One. The study presents high quality and deals with important clinical issue, such type of study is needed. I have only few small remarks that authors should address properly.

I recommend to accept the manuscript after minor revision.

There are only some points to correct:

- please provide the list of abbreviations

- introduction and discussion section need improvement – please provide information on how your results will translate into clinical practice

- in discussion section please provide study strong points and study limitation section

- please correct typos

I recommend to accept the manuscript after minor revision.

6. PLOS authors have the option to publish the peer review history of their article (what does this mean?). If published, this will include your full peer review and any attached files.

Reviewer #1: No

Reviewer #2: No

---

## [Author Response · Author response to Decision Letter 0]

30 Oct 2020

Article: Bed-side measures for diagnosis of low muscle mass, sarcopenia, obesity, and sarcopenic obesity in patients with chronic kidney disease under non-dialysis-dependent, dialysis dependent and kidney transplant therapy

Response to Reviewers

JOURNAL REQUIREMENTS

AUTHORS. OK

2. In your Methods section, please provide additional information about the participant recruitment method and the demographic details of your participants. Please ensure you have provided sufficient details to replicate the analyses such as:

a) the recruitment date range (month and year)

AUTHORS. Included in methodology section: “(…) from May 2017 to May 2019”.

b) any sample size calculations

AUTHORS. Included in methodology section: “Sample size was calculated based on sarcopenia prevalence in CKD, assuming an expected prevalence of 5%, with estimation of at least 80 patients for each CKD treatment”.

c) a description of how participants were recruited 

AUTHORS. Included in methodology section: “The Ribeirão Preto Medical School Ethics Committee approved the study (protocol number 2053045). All patients were invited by the researcher and those interested in participating the study read and signed the informed consent form before the procedures began”.

d) descriptions of where participants were recruited and where the research took place

AUTHORS. Included in methodology section : “(…) at a tertiary care hospital, the University Hospital of the Ribeirão Preto Medical School and at a dialysis clinic, the Nephrology Service of Ribeirão Preto”.

AUTHORS. OK

4. Please include captions for your Supporting Information files at the end of your manuscript, and update any in-text citations to match accordingly. Please see our Supporting Information guidelines for more information:

http://journals.plos.org/plosone/s/supporting-information.

AUTHORS. OK

5. Please ensure that you include a title page within your main document. You should list all authors and all affiliations as per our author instructions and clearly indicate the corresponding author.

AUTHORS. OK

6. Please amend your manuscript to include your abstract after the title page.

AUTHORS. OK

REVIEW COMMENTS TO THE AUTHOR

REVIEWER #1

This is a cross-sectional study (n 265), followed by a prospective follow-up study with 33% (n 87) of the patients initially included. The study provides a detailed description of the prevalence of obesity, sarcopenia, and sarcopenic obesity in a cohort of patients with CKD (CKD Stages 1 to 3a, PD, hemodialysis, and kidney transplantation). The paper provides relevant information comparing bed-sides measures with dual-energy X-ray absorptiometry in a crosssectional study in CKD patients. It also tries to establish cut-off values for CKD patients. 

However, the diagnostic capabilities of the measurements of muscle and fat mass are analyzed mainly according to sex and without considering the different situations of renal function; this makes it difficult to assess the sensitivity and specificity of the measures in different "renal situations" and also questions the validity of the cut-off points. An analysis of the diagnostic capacity of the tests in relation to the renal situation is necessary.

AUTHORS. We performed additional analyzes with stratification of the sample by the CKD treatment subgroups. The results are presented in the table S4. 

We add descriptions of this analysis in results and discussion sections.

Results section. “Data about sensitivity and specificity of the best bed-side measurements for low muscle mass (AFFM and CC), obesity (FMIBCM and WC/H), sarcopenia (AFFM+HGS and CC+HGS) and sarcopenic obesity (FMIBCM+AFFM) in each CKD subgroups are presented in S4 Table. The analysis stratified by CKD group showed wide 95%IC for diagnosis of sarcopenia and sarcopenic obesity (S4 Table)”.

Discussion section. “Analysis with stratification of CKD subgroups showed good performance of AFFM, CC, FMIBCM and WC/H for all CKD groups, mainly for BIS measurements. On the other hand, although a good sensibility and specificity, the wide 95%CI of sarcopenia and sarcopenic obesity diagnosis by bed-side measurements in each CKD group show a need of validation in a larger sample size before routine use of these measurements”. 

In the prospective study (n 87), the renal situation of the patients is not clear. A graph comparing the initial situation and follow-up according to the initial classification (renal and sarcopenic-obesity) would also be interesting. 

AUTHORS. We performed a follow-up graph presented in figure S1. Additional analyzes with comparison of data between group 1 (patients that were evaluated only in the first assessment) and group 2 (patients that were evaluated in the first and second assessment) were also done. We add descriptions of this analysis in results section.

Results. “These results are supported by Figure S1, in which it is possible to observe that there was a tendency for patients with normal body composition in the first evaluation to be diagnosed with a worse body composition in the second evaluation; changes to a better body composition occurred more rarely”.

“As only few patients participated in the second assessment, to evaluate if there was some selection bias that could influenced the direction of body composition changes, we compared cross-sectional data from patients that were evaluated only in the first assessment with the ones that participated in the first and second assessment (S2 Table). No statistical difference was found between groups.”

The study excluded patients over 60 years of age, a group that constitutes an important and growing population with a high prevalence of CKD, sarcopenia, and obesity. This point should be discussed more extensively in the discussion. 

AUTHORS. We add additional information in Limitations section. “Sex and age are well known important factors in determine skeletal muscle mass and body adiposity. In addition, physical function and eGFR declines with age. So, our mayor study limitation is the exclusion of patients over 60 years of age. As we evaluated only adult CKD patients, our results cannot be applied to the elderly. Validation of the measures and cut-off points in elderly are needed not only because of differences in body composition but also in the accuracy of instruments (…)”.

Minor review 

Abstract: the acronyms DXA and HGS must be defined in the abstract on its first use.

AUTHORS. OK.

It would be interesting to include a table of the different measures according to the renal situation.

AUTHORS. As the article already have so many tables and figures (6 tables in the text article and 4 as supplementary material; 5 figures as supplementary material), we provided information about CKD patients according to renal situation throughout the text, more precisely in the first, second and third paragraph of results section. 

Page 6, lines 6 and 7, the sentence is understood, but it isn't very clear. 

AUTHORS. We were unable to find the sentence. If it was about the comparison between renal groups, we tried to write in a better form.

REVIEWER #2

I was honored to review the manuscript entitled “Bed-side measures for diagnosis of low muscle mass, sarcopenia, obesity, and sarcopenic obesity in patients with chronic kidney disease under non-dialysis dependent, dialysis dependent and kidney transplant therapy” submitted to Plos One. 

The study presents high quality and deals with important clinical issue, such type of study is needed. I have only few small remarks that authors should address properly. I recommend to accept the manuscript after minor revision. 

There are only some points to correct: 

- please provide the list of abbreviations 

AUTHORS. We provided a list of abbreviations in title page.

- introduction and discussion section need improvement – please provide information on how your results will translate into clinical practice 

AUTHORS. We add additional information in the Introduction section. “(…) As negative changes of body composition and nutritional status significantly increase morbidity and mortality risk in CKD patients, the early diagnosis of such changes is of fundamental importance. However, DXA availability is restricted, usually applied in diagnostic studies and rarely feasible in clinical practice. Therefore, the use of tools that are more easily available in routine nutritional assessment, such as anthropometry and bioelectrical impedance analyses, could help in the early identification of nutritional status impairment, improving clinical outcomes by early interventions(…)”.

Additional information was also provided in the Discussion section. “(…) Given the well-known strong association of poor nutritional status with a worse clinical prognosis and higher mortality, nutritional assessment has to be done frequently and as part of the regular CKD patient care. In order to make an early identification allowing interventions and improving clinical outcomes, tools with diagnostic capacity that are easily available are needed. We revealed that BIS and anthropometric measurements, such as AFFM and CC, could be used with the proposed cutoffs for low muscle mass diagnosis, and with the combination of HGS, for sarcopenia diagnosis. We also specified cutoffs for FMIBCM and WC/H for obesity and FMIBCM+AFFM for sarcopenic obesity diagnosis. Training of multiprofessional teams, however, is mandatory to carry out these simple measures that can be used routinely even in institutions that do not have high-cost equipment. Cross-validation in a large sample size with participation of four CKD treatment groups is still needed”. 

- in discussion section please provide study strong points and study limitation section 

AUTHORS. We add a strengths and Limitations sections.

Strengths. “We evaluated not only one but four conditions related to impairment of nutritional status, as low muscle mass, sarcopenia, obesity and sarcopenic obesity. Also, we have a large sample of CKD patients under the four type of treatment, as NDD, HD, PD and KTx. In addition, we applied the new definition of sarcopenia by EWGSOP with DXA analysis, a reference method for body composition evaluation, in all patients. We conducted a cross sectional and prospective study with important data of body composition trajectory in CKD. Thorough statistics were carried out allowing a comprehensive and in-depth analysis of our results. Finally, we suggest easily available anthropometric and bioelectrical impedance measures as tools for diagnosis of the conditions previously cited”.

Limitations. “Sex and age are well known important factors in determine skeletal muscle mass and body adiposity. In addition, physical function and eGFR declines with age. So, our mayor study limitation is the exclusion of patients over 60 years of age. As we evaluated only adult CKD patients, our results cannot be applied to the elderly. Validation of the measures and cut-off points in elderly are needed not only because of differences in body composition but also in the accuracy of instruments. Also, longer prospective evaluations could provide more information about the body composition trajectory of CKD patients and associations between outcomes and mortality. The PD group was underrepresented, which may have affected CKD subgroup analysis. Moreover, we applied a definition of sarcopenic obesity that takes into account only low muscle mass and obesity, not including muscle function, such as low HGS. Although there is not a consensus on sarcopenic obesity definition, which varies considerably, we think this information should be considered. Finally, the sample included only 2 patients with pre-sarcopenic obesity in the prospective assessment, which limited our analysis of body composition changes in this group”.

- please correct typos I recommend to accept the manuscript after minor revision.

AUTHORS. OK

---

## [Decision Letter · Decision Letter 1]

9 Nov 2020

Bed-side measures for diagnosis of low muscle mass, sarcopenia, obesity, and sarcopenic obesity in patients with chronic kidney disease under non-dialysis-dependent, dialysis dependent and kidney transplant therapy

PONE-D-20-28011R1

Dear Dr. Bellafronte,

We’re pleased to inform you that your manuscript has been judged scientifically suitable for publication and will be formally accepted for publication once it meets all outstanding technical requirements.

Kind regards,

Mauro Lombardo

Academic Editor

PLOS ONE

Additional Editor Comments (optional):

Reviewers' comments:

Reviewer's Responses to Questions

**Comments to the Author**

1. If the authors have adequately addressed your comments raised in a previous round of review and you feel that this manuscript is now acceptable for publication, you may indicate that here to bypass the “Comments to the Author” section, enter your conflict of interest statement in the “Confidential to Editor” section, and submit your "Accept" recommendation.

Reviewer #1: All comments have been addressed

Reviewer #2: All comments have been addressed

2. Is the manuscript technically sound, and do the data support the conclusions?

Reviewer #1: Yes

Reviewer #2: Yes

3. Has the statistical analysis been performed appropriately and rigorously? 

Reviewer #1: Yes

Reviewer #2: Yes

4. Have the authors made all data underlying the findings in their manuscript fully available?

Reviewer #1: Yes

Reviewer #2: Yes

5. Is the manuscript presented in an intelligible fashion and written in standard English?

Reviewer #1: Yes

Reviewer #2: Yes

6. Review Comments to the Author

Reviewer #1: (No Response)

Reviewer #2: All comments have been addressed. All changes have been implemented. I recommend accepting the manuscript.

The manuscript is presented in an intelligible fashion and is written in standard English.

7. PLOS authors have the option to publish the peer review history of their article (what does this mean?). If published, this will include your full peer review and any attached files.

Reviewer #1: No

Reviewer #2: No

---

## [Editor Report · Acceptance letter]

12 Nov 2020

PONE-D-20-28011R1 

Bed-side measures for diagnosis of low muscle mass, sarcopenia, obesity, and sarcopenic obesity in patients with chronic kidney disease under non-dialysis-dependent, dialysis dependent and kidney transplant therapy 

Dear Dr. Bellafronte:

I'm pleased to inform you that your manuscript has been deemed suitable for publication in PLOS ONE. Congratulations! Your manuscript is now with our production department. 

Kind regards, 

on behalf of

Dr. Mauro Lombardo 

Academic Editor

PLOS ONE